# Prolactin at moderately increased levels confers a neuroprotective effect in non-secreting pituitary macroadenomas

David A. Paul[1,☯], Alejandra Rodrigue[2,3☯], Nicholas Contento[2], Sam Haber[2,4], Ricky Hoang[2], Redi Rahmani[1], Adnan Hirad[2], Ismat Shafiq[5], Zoë Williams[1,3,6], G. Edward Vates[1]*

1 Department of Neurosurgery, University of Rochester Medical Center, Rochester, New York, United States of America, 2 University of Rochester School of Medicine and Dentistry, Rochester, New York, United States of America, 3 Department of Ophthalmology, University of Rochester Medical Center, Rochester, New York, United States of America, 4 Department of Brain and Cognitive Sciences, University of Rochester, Rochester, New York, United States of America, 5 Division of Endocrinology and Metabolism, University of Rochester Medical Center, Rochester, New York, United States of America, 6 Department of Neurology, University of Rochester Medical Center, Rochester, New York, United States of America

☯ These authors contributed equally to this work.
* Edward_vates@urmc.rochester.edu

**Data Availability Statement:** The deidentified dataset can be found in the Figshare repository. DOI: 10.6084/m9.figshare.19678953.

## Abstract

### Context

Prolactin, a hormone synthesized by the anterior pituitary gland demonstrates promise as a neuroprotective agent, however, its role in humans and in vivo during injury is not fully understood.

### Objective

To investigate whether elevated levels of prolactin attenuate injury to the retinal nerve fiber layer (RNFL) following compression of the optic chiasm in patients with a prolactin secreting pituitary macroadenoma (i.e., prolactinoma).

### Design setting and participants

A retrospective cross-sectional study of all pituitary macroadenoma patients treated at a single institution between 2009 and 2019.

### Main outcome measure(s)

Primary outcome measures included RNFL thickness, mean deviation, and prolactin levels for both prolactin-secreting and non-secreting pituitary macroadenoma patients.

### Results

Sixty-six patients met inclusion criteria for this study (14 prolactin-secreting and 52 non-secreting macroadenoma patients). Of 52 non-secreting macroadenoma patients, 12 had moderate elevation of prolactin secondary to stalk effect. Patients with moderate elevation

**Funding:** DP was supported by a UL1 TR002001 grant via the NIH/National Center for Advancing Translational Sciences at the University of Rochester and resources provided by the Program for Translational Brain Mapping at the University of Rochester. The program for Translational Brain Mapping was established, in part, with support from Norman and Arlene Leenhouts (see: www.tbm.urmc.rochester.edu). AR was supported by a CTSI TL1 TR000096 grant via the National Center for Advancing Translational Sciences at the University of Rochester.

**Competing interests:** The authors have declared that no competing interests exist.

in prolactin demonstrated increased RNFL thickness compared to patients with normal prolactin levels (p < 0.01). Additionally, a significant positive relation between increasing levels of prolactin and RNFL thickness was identified in patients with moderate prolactin elevation (R = 0.51, p-value = 0.035). No significant difference was identified between prolactinoma patients and those with normal prolactin levels.

## Conclusions

Moderately increased serum prolactin is associated with increased RNFL thickness when compared to controls. These associations are lost when serum prolactin is < 30 ng/ml or elevated in prolactinomas. This suggests a neuroprotective effect of prolactin at moderately increased levels in preserving retinal function during optic chiasm compression.

## Introduction

Pituitary adenomas account for approximately 15% of all intracranial tumors [1]. These tumors can cause hormonal derangements by either the over- or under-production of pituitary hormones. If the tumor is sufficiently large, patients can present with a stereotyped vision loss secondary to tumor impingement on the optic nerves, chiasm or tract [1, 2]; and prolonged compression of these structures leads to a characteristic thinning of the retinal nerve fiber layer (RNFL) [3]. For many patients, decompression of the anterior visual pathway by tumor removal (with surgery) or tumor shrinkage (with medication) can result in rapid visual recovery–although the factors that predict which patients experience recovery and to what extent they will recover remains an active area of investigation. RNFL thickness–a surrogate for retinal ganglion cell axonal health–has been used as a biomarker for indexing various mechanisms of delayed axonal degeneration [4], including white matter injury following anterior visual pathway compression [5–7]. Here, we investigate the relation between serum hormone levels of prolactin, RNFL thickness and visual function in a retrospective cohort of pituitary macroadenoma patients.

Prolactin (PRL), a hormone synthesized in the anterior pituitary gland and associated with lactation, also demonstrates diverse physiologic functions, including processes that mediate neuroprotection. Prolactin has been implicated in oligodendrocyte progenitor cell proliferation [8], neurotrophic factor release [9], and increased white matter volume [10]. Together, these studies highlight the various neuroprotective roles of PRL; yet PRL's role in humans and *in vivo* during injury is not fully understood. By studying the effects of compression on visual pathway structures, we can explore the relation between increasing PRL levels and retinal ganglion cell axonal health.

Prolactin can be elevated one of two ways: 1) excess production from lactotroph cells secondary to growth of a prolactinoma, or 2) a decrease in the inhibition of prolactin secretion resulting from a physiologic block of dopamine delivery from the hypothalamus through the pituitary stalk–a phenomenon known as "stalk effect". This phenomenon is seen with mass effect from non-secreting macroadenomas. [11, 12]

The present study identified two groups for investigation: patients with macro-prolactinomas (PRO), and control patients with non-secreting (NS) macroadenomas. The control group was further subdivided into those non-secreting macroadenomas with hyperprolactinemia (NS+) from stalk effect, and those non-secreting macroadenoma cases without hyperprolactinemia (NS). This natural variation in serum prolactin allows for the study of varying levels of

prolactin on measures of retinal ganglion cell axonal integrity and visual function. We hypothesized that increased serum levels of PRL are associated with a decrease in secondary injury from compression as measured by RNFL thickness.

## Materials and methods

All patients with pituitary tumors treated at the University of Rochester Multidisciplinary Pituitary Clinic between 2009 and 2019 were evaluated for inclusion in this study. Inclusion criteria for both the prolactinoma and control groups were age greater than 18, pituitary tumor >1 cm in any dimension (i.e., macroadenoma), serum prolactin level recorded at time of diagnosis, and ophthalmologic testing performed at or after diagnosis. Both Male and Female patients were included in the study, as defined by their biological sex on chart review. Macroadenomas were defined based on volume, independent of the cell of origin or serum prolactin level as is standard in the literature [13]. Of 2,728 patients treated at our institution (including both micro and macroadenoma patients), 239 were identified to have both Humphrey 24–2 perimetry and spectral domain optical coherence tomography (SD-OCT), which measures the peripapillary retinal nerve fiber layer. Of those patients, 66 were macroadenoma patients. Control patients were chosen from the population of non-secreting pituitary tumor patients. The study protocol was approved by the institutional review board of the University of Rochester and the need for consent was waived. All data were anonymized before being accessed.

### Power analysis

A power analysis was conducted based on the initial hypothesis that elevated levels of prolactin in prolactin-secreting macroadenoma patients will demonstrate attenuation of injury to the retinal nerve fiber layer compared with non-secreting macroadenoma patients. Previous research indicates that pre-operative mean RNFL thickness in pituitary macroadenoma patients (standard deviation) is 81.9 μm (8.8) [14]. We anticipated a similar mean RNFL thickness for non-secreting pituitary macroadenoma patients in this study. For patients with elevated prolactin, we anticipated a mean RNFL thickness to be closer to the control group mean of 91.9 μm (9.7), as reported in Moon et al. [14]. A total sample size of 30 participants (15 per group) was found to provide 80% power to detect a 10 μm difference in mean RNFL thickness between the two groups, using a Welch's T-test and a 5% significance level.

### Tumor characteristics

All patients had pituitary tumors greater than 1 cm in any direction. Tumor size was identified by a single trained investigator and the largest dimension in any direction was recorded. Tumors not originating in the pituitary gland were excluded.

### Measurement of serum prolactin levels

Serum prolactin was measured as part of the routine clinical workup for each patient via FDA approved Roche Elecsys Prolactin II Assay (Electrochemiluminescence Immunoassay [ECLIA]; Roche Diagnostics; Indianapolis, IN) with a reference range of 4.8–23.3ng/ml [15]. This assay demonstrates increased sensitivity to detect the concentration of monomeric prolactin by avoiding false elevation secondary to reactivity with macroprolactin [16]. Our clinical laboratory is a Clinical Laboratory Improvement Amendments (CLIA) certified laboratory and accredited by the College of American Pathologists and New York State Department of Health. Briefly–antigen-specific monoclonal antibodies were coated onto beads and mixed with the PRL sample to allow an immune reaction to occur. Unbound sample was then washed

away and a second monoclonal antibody with an electrochemiluminescent probe was added to the mixture to bind the PRL-antibody complex. An electrode was then introduced to the sample, which generates quantifiable electrochemiluminescence via an oxidation-reduction reaction that directly correlates to the amount of PRL present. All data for this study were recorded prior to any treatments including dopamine agonists and/or surgery.

## Ophthalmologic data

All included subjects had ophthalmologic examination data which included RNFL evaluation with SD-OCT–optic disc cube 200 × 200 protocol as described previously [7]. Measures of RNFL thickness were reported as a function of clock hour position of the fovea and subsequently grouped into anatomical quadrants based on a standard division of the visual field (e.g., for right eyes; superior quadrant: 11 and 1 o'clock; nasal quadrant: 2, 3, and 4 o' clock; inferior quadrant: 5 and 7 o' clock; and the temporal quadrant: 8, 9, and 10 o'clock). The 12 o'clock and 6 o'clock positions were excluded from analysis secondary to nontemporal overlap of retinal ganglion cell projections [17]. The temporal RNFL quadrant demonstrates increased sensitivity to injury with pituitary tumors secondary to its association with crossing retinofugal fibers at the level of the optic chiasm [14]. Average peripapillary RNFL values were obtained for each eye and within each quadrant (prolactinoma group n eyes = 27; control group n eyes = 104). One subject from the prolactinoma group had RNFL data only from one eye. Automated perimetry testing was performed using Humphrey Field Analyzer (24–2 SITA-Standard algorithm); Carl Zeiss Meditec, Dublin, CA, USA. Visual function was assessed using mean deviation–recorded from the Humphrey automated perimetry test for 27 eyes in the prolactinoma group and 104 eyes in the control group.

## Statistical analysis

Statistical analysis was performed using R (version 3.6.1). Normality was assessed using the Shapiro-Wilks test. Parametric data were analyzed using one-tailed Welch's *t*-test where the alternative hypothesis stated that the prolactinoma group's retinal layer measurements would be greater than the controls. Eyes were treated independently under the assumption that compressive injury may not be equal in both eyes, thus PRL's assumed effects may also vary in each eye. Nonparametric data were analyzed using the Mann–Whitney U test. A one-way ANOVA was used to assess differences in average RNFL thickness between prolactinoma patients, non-secreting controls with elevated PRL due to stalk effect (NS+), and controls with normal PRL (NS). Pearson linear regression was used to analyze mean deviation and prolactin levels for each group. Mahalanobis distance was calculated to identify outliers in the mean deviation data. Multiple linear regression analysis was performed to assess the effects of PRL, age, and tumor volume on RNFL thickness in the PRO group. A $p$-value $<0.05$ was considered statistically significant.

## Results

Out of the 239 patients who were identified to have formal ophthalmologic testing, a total of 66 met inclusion criteria for the study. This includes 14 prolactinoma patients and 52 non-secreting pituitary tumor patients. Of the non-secreting pituitary tumors, 12 were identified to have stalk effect, and 40 patients had normal levels of serum prolactin. See Fig 1.

There was no significant difference in biological sex between patients in the PRO and Control groups (Table 1). Age at diagnosis for the PRO (43.2 ± 18.8 years) was significantly younger than the Control group (59.0 ± 15.0 years) (p-value = 0.0076). Median serum prolactin levels for the prolactinoma group (744.1 ng/ml) were significantly greater than the controls

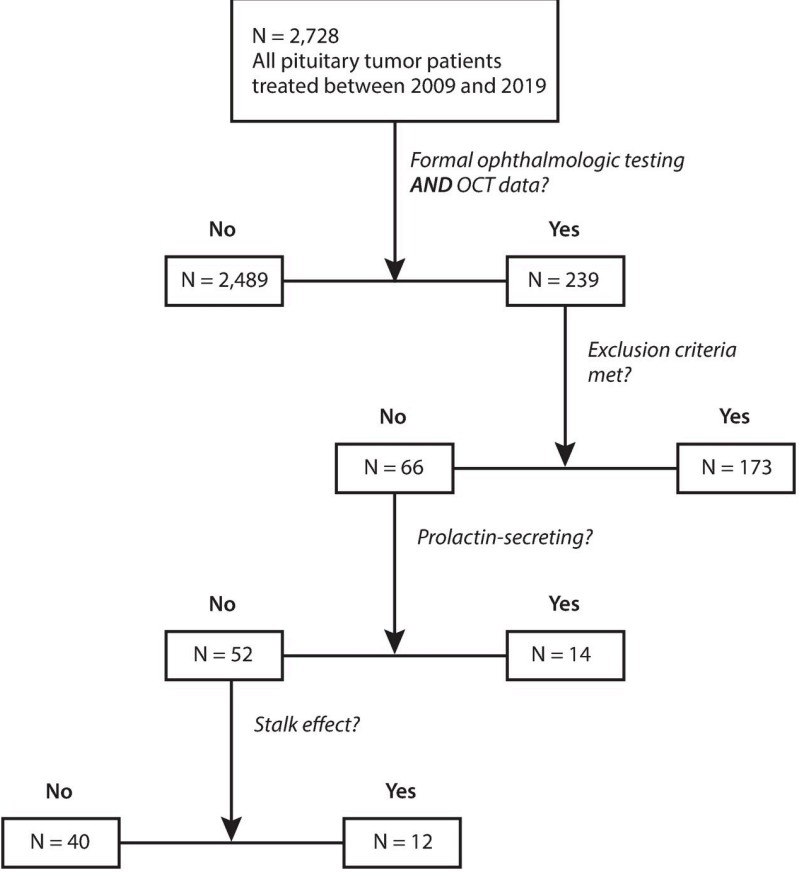

**Fig 1. The patient selection flow chart.** Abbreviations: OCT, optical coherence tomography.

**Table 1. Characteristics of the study population including mean deviation recorded by Humphrey 24–2 automated perimetry.** For mean deviation PRO n = 22 eyes; Controls n = 51 eyes. Welch's T-test was used for normally distributed data, Mann-Whitney U test was performed for non-normally distributed data, and Chi-squared was used for nominal data.

| | PRO Group | Controls | P-value |
|---|---|---|---|
| | Mean (SD) | Mean (SD) | |
| Participants, n | 14 | 52 | |
| Eyes, n | 27 | 104 | |
| Age at diagnosis | 43.2 (18.8) | 59.0 (15.0) | 0.0076 |
| Male/Female | 9/5 | 29/23 | 0.33 |
| Prolactin at diagnosis (ng/ml) [†] | 744.1 | 14.2 | <.00001 |
| Mean deviation (dB)[††] | -4.4 (4.2) | -6.9 (8.0) | 0.79 |
| Tumor Size (mm) | 22.1 (14.7) | 25.1 (8.9) | 0.20 |

OCT = optical coherence tomography;

[†] Median value reported;

[††] different n

**Table 2. RNFL thickness measurements recorded from SD-OCT optic disc and macular cube scan protocols.**
One-sided Welch T-test was used to determine significance. The data demonstrate a statistically significant difference in RNFL thickness between PRO and control groups in the temporal quadrant.

| | | PRO Group | Controls |
|---|---|---|---|
| | | Mean (SD) | Mean (SD) |
| RNFL Eyes, n | | 27 | 104 |
| RNFL Average (µm) | | 82.7 (15.6) | 81.5 (19.5) |
| RNLF Quadrants (µm) | Superior | 104.4 (23.1) | 102.3 (25.1) |
| | Nasal | 63.7 (10.9) | 65.2 (13.3) |
| | Temporal | 57.1 (10.8) * | 52.1 (14.0) |
| | Inferior | 105.4 (26.7) | 105.7 (27.3) |

RNFL = retinal nerve fiber layer;

*p-value <0.05

(14.2 ng/ml) (p-value < 0.00001) (Table 1). There was no significant difference in tumor size between the PRO (22.1 mm ± 14.7) and control groups (25.1 ± 8.9; p-value = 0.20). See Table 1. Of the patients in the PRO group, 22 eyes underwent visual field testing with mean deviation measurements. Of the Control group 51 eyes had mean deviation data available from visual field testing. Ophthalmologic data revealed that there was no difference between mean deviation of the PRO (-4.4 ± 4.2 dB) and control groups (-6.9 ± 8.0 dB) (*p*-value = 0.79) (Table 2). Average RNFL thickness was similar in both groups (Table 2), however, the temporal quadrant of RNFL was significantly thinner in the control group than the prolactinoma group (*p*-value = 0.04).

Further analysis of RNFL, assessing the relation between PRO patients and both NS (n = 80 eyes, mean PRL = 13.1 ng/ml) and NS+ (n = 24 eyes, mean PRL = 50.5 ng/ml) patients, revealed a significant difference between the groups (p-value = 0.0027)–See Table 3. *Post-hoc* analysis also demonstrated a significant difference between RNFL thickness of NS+ and NS (p-value = 0.0018). See Fig 2. This analysis was repeated for tumor size, which identified no significant differences between the groups. In plotting the mean deviation data, a single data point varied from the rest. Prior to running statistical analysis on the data set, Mahalanobis distance was used–which identified an extreme outlier in the mean deviation data for the NS + group (See S1 Fig). This was validated post-hoc using the extreme studentized deviate test

**Table 3. Subject characteristics by group including visual ability as mean deviation recorded by Humphrey 24–2 automated perimetry (PRO n eyes = 22, controls NS+ n eyes = 18, controls NS n eyes = 33) and SD-OCT average RNFL thickness (PRO n eyes = 27, controls NS+ n eyes = 24, and NS n eyes = 80).** One way ANOVA was used to determine significance and Chi-squared was used for nominal data.

| | PRO | Controls | | p- value |
|---|---|---|---|---|
| | | NS+ | NS | |
| Subjects, n | 14 | 12 | 40 | |
| Males/Females | 9/5 | 5/ 7 | 24/16 | 0.45 |
| Age at diagnosis | 43.2 (18.8) | 55.3 (18.6) | 60.2 (13.8) | 0.0044 |
| Prolactin (ng/ml) | 2453.3 (3907.6) | 50.5 (21) | 13.1 (7.2) | 0.00013 |
| Tumor size (mm) | 22.1 (14.7) | 26.4 (7.6) | 24.6 (10.0) | 0.55 |
| Mean deviation (dB) | -4.4 (4.2) | -4.4 (6.1) | -7.1 (8.7) | 0.28 |
| Average RNFL Thickness (um) | 82.7 (15.6) | 91.7 (19.1) | 78.4 (15.6) | 0.0027 |

OCT = optical coherence tomography; value reported as mean (st. deviation)

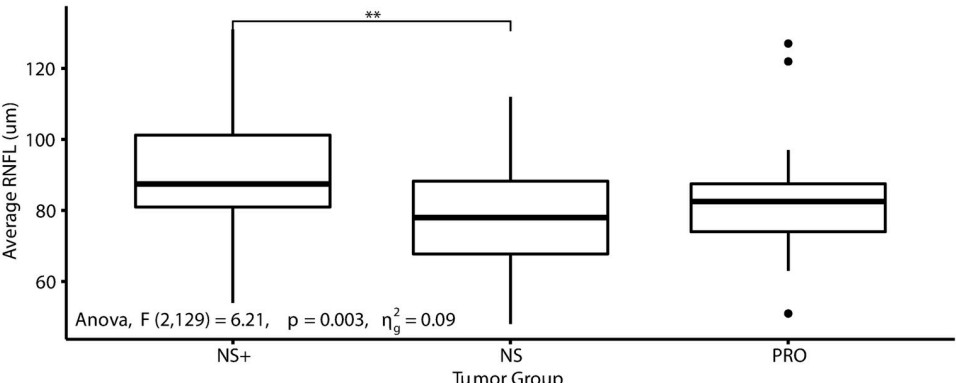

**Fig 2. Average RNFL distribution of PRO and controls which were further divided into those with hyperprolactinemia as determined clinically as > 30 ng/ml (NS+) and those with normal prolactin (NS).** For RNFL analysis PRO eyes n = 27, NS+ eyes n = 24, and NS eyes n = 80. RNFL did differ significantly between the groups (p-value 0.0027) using one-way ANOVA. Post Hoc (Tukey) analysis demonstrated significant difference between the NS+ and NS groups (p-value 0.0018). **p-value < 0.01.

(z-score 3.66, p <0.05). Mahalanobis distance and extreme studentized deviate test did not identify any other outliers in either the PRO or NS groups. The extreme outlier was removed from the mean deviation analysis.

Pearson correlation analysis of prolactin as a function of mean deviation revealed little to no association in the PRO group (R = -0.23, $R^2$ = 0.051, p-value = 0.31). However, among control participants subdivided by stalk effect, a significant inverse correlation with mean deviation was identified for the NS group (R = -0.47, $R^2$ = 0.23, p-value = 0.0053) and a significant direct correlation for the NS+ group (R = 0.51, $R^2$ = 0.26, p-value = 0.035). See Fig 3.

## Discussion

In this study, we demonstrate a positive relation between prolactin and RNFL thickness during anterior visual pathway injury from pituitary masses. This trend was significant when assessing

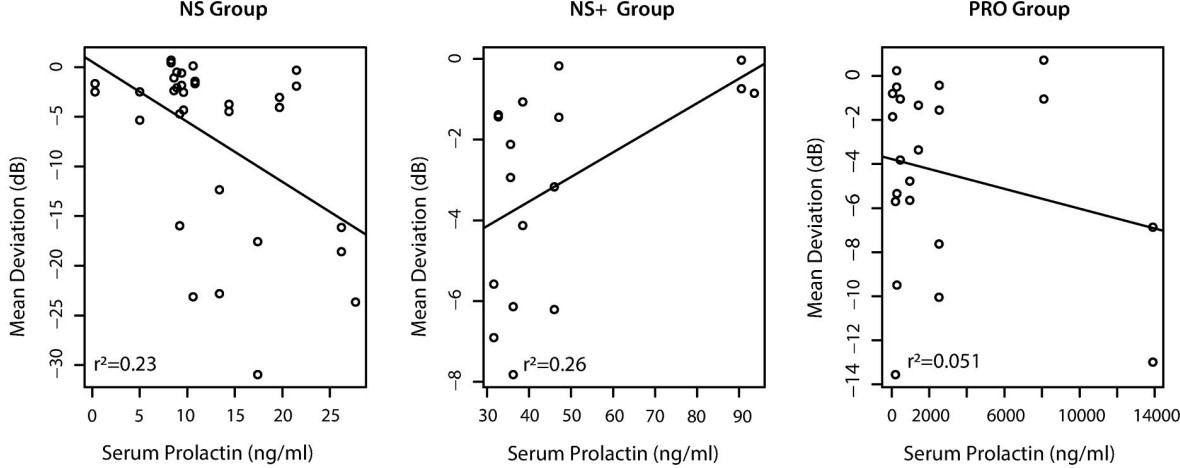

**Fig 3. Pearson linear regression analysis was performed for all subjects with the controls divided by subgroups (NS+ and NS).** Analysis of PRL as a function of mean deviation in the PRO group revealed little negative association (R = -0.23, p-value = 0.31). When analyzing the controls by subgroups, NS+ demonstrated a significant correlation between mean deviation and serum PRL level (R = 0.51, p-value = 0.035) while the NS group demonstrated an inverse correlation that approached significance (R = -0.47, p-value = 0.0053).

PRL levels and RNFL for non-secreting pituitary tumors with moderately elevated PRL levels (NS+) due to stalk effect. We also demonstrate a significant relation between serum PRL and mean deviation. Together, these data suggest that at moderately elevated levels, PRL may confer neuroprotection against injury with preservation of RNFL thickness.

It is also important to note that in this study we focused on the relation between prolactin and RNFL thickness at the time of diagnosis–an approach that is agnostic to treatment type (e.g. open vs trans-sphenoidal tumor resection or medical therapy). This cross-sectional approach removes confounding factors related to the potential effects of treatment on both RNFL thickness and visual function, and the role that decompression plays in mediating recovery. Thus, an important distinction is that our data demonstrate the potential for preservation of RNFL thickness as opposed to facilitating recovery. Further research is necessary to determine the impact of serum prolactin on neural recovery after injury–which might inform therapeutic approaches for central nervous system pathologies other than pituitary tumors.

Our study is novel in that we assess the effects of elevated PRL in two hyperprolactinemia states; non-secreting tumors that cause stalk effect and moderately elevated PRL levels, and prolactinoma patients with extremely elevated PRL levels. One previous study investigating changes in vision as a function of hormone status in patients with pituitary macroadenomas focused primarily on medically treated functional pituitary tumors only and demonstrated no significant relationship in 6 patients [18]. Hyperprolactinemia from stalk effect was not considered.

One limitation of the current study is the small sample size. Clinical use of OCT at our institution is limited to patients who experience "vision loss" as their chief complaint. Additionally, as one of the largest regional referral centers for pituitary tumor patients in the United States [19], many patients complete their initial ophthalmologic workup outside of our academic medical center. Our findings, specifically with respect to the relation between moderate elevation in PRL and RNFL thickness warrants further study both in a larger cohort of patients and in a prospective fashion.

PRL has been studied for its neuroprotective role in the retina and in white matter. In the retina, PRL receptors have been identified in the retinal pigment epithelium and found to be protective against cell death from oxidative stress [20]. Additional evidence has identified PRL as a trophic factor that regulates glial-neuronal interactions and protects against retinal degeneration [21]. Whether these mechanisms are triggered by PRL during compressive injury has not been elucidated, but provides a possible route for the protective characteristics observed in this study. PRL's role in white matter injury has been studied most recently in multiple sclerosis patients and in mouse models of spinal cord injury. These studies attribute increased oligodendrocyte proliferation and remyelination after injury to increased PRL levels [8] and increased white matter volume [10]. We have previously shown that remyelination is possible after compressive injuries to the anterior visual pathway [22] and that measures of diffusion known to correlate with myelination are sensitive to varying levels of serum prolactin in a patient with empty-sella syndrome [7]. Due to PRL's role in remyelination, this provides another possible mechanism for its protective quality following compressive injuries.

OCT derived measurements of RNFL thickness have been used extensively to diagnose and follow various pathologies of the visual system including compressive injuries [5, 6, 14, 23]. Pre-operative RNFL measurements have also been shown to predict post-operative visual outcomes for pituitary tumor patients [6]. These measurements may serve as biomarkers for injury along the visual pathway where thinning indicates ongoing injury and disease progression [7, 24]. This allows for RNFL measurements to be used to quantify the efficacy of neuroprotective agents, like PRL. Our results demonstrate that moderately increased serum PRL levels are associated with increased RNFL thickness and mean deviation.

We demonstrated that moderately elevated PRL levels like those found in patients with stalk effect were more strongly correlated with RNFL thickness whereas the extremely elevated levels of PRL in the PRO group conferred no preservation of RNFL thickness. Notably, the control group in our dataset was statistically older than the PRL group. Given that RNFL thickness has been shown to decrease slowly with age it may be expected for the controls whose average age was greater than the prolactinoma group to have thinner retinas [25, 26]. Additionally, a 24-hour PRL collection study found that older patients had a lower pulse mass and lower peak values of PRL secreted [27], although average PRL values overall did not significantly decline with age. Taken together, our data demonstrates that even with the added variable of age there is a preservation of retinal thickness in the group with mildly elevated prolactin. In other words, regardless of the natural retinal thinning and PRL changes that may come with age, older patients with moderately elevated PRL had greater RNFL thickness than patients with significant hyperprolactinemia. In the NS group, PRL was inversely associated, suggesting that PRL at these lower levels may also result in worsening visual function during injury. Additional studies of visual function are needed to investigate this possibility.

When analyzing mean deviation, the PRO group demonstrated little to no association with PRL. While there was no observed increase in visual function (as measured by mean deviation) at increasingly higher levels of prolactin in the PRO group, this does not exclude the possibility that elevated PRO prevents further decline in visual function. This may also indicate that PRL is protective within a certain range. Physiologically PRL demonstrates both inhibitory and excitatory actions. In the hypothalamus, elevated PRL levels inhibit the secretion of gonadotrophin-releasing hormone. Further investigation is required to elucidate whether a similar mechanism may be involved in prolactin's neuroprotective function. Alternatively, chronic, extremely elevated levels of PRL in the PRO group can cause receptor desensitization and downregulation thus preventing the neuroprotective potential. The kinetics of the human prolactin receptor have shown it to behave with its agonist in a bell-shaped fashion implying supersaturation at high levels and decreased pathway activation at low levels [28]. Thus, the moderately elevated levels of PRL may work at peak PRL receptor activity without desensitization. Additionally, proteolytic cleavage of PRL generates active peptides (vasoinhibins/16K PRL)–which have been shown to have effects on vasculature by promoting vasopressin release, and on neurons by inhibiting neurite outgrowth [29]. High levels of PRL would in turn result in increased levels of vasoinhibins, whose function may prevent or oppose the neuroprotective action of PRL.

In summary, this study demonstrates attenuation of injury to the retina during anterior visual pathway injury with moderately elevated prolactin levels in non-secreting pituitary tumor patients. We show that hormone status, specifically prolactin, a known neuroprotective agent, may influence the structure-function relationship of the visual system during injury. Additional studies with larger sample sizes that include visual function tests could further elucidate the role of prolactin in preserving function after injury.

## Supporting information

**S1 Fig. Outlier identification was determined using Mahalanobis distance. A)** Classic Mahalanobis distance is plotted for each data point of the NS+ group mean deviation with a threshold of 3. **B)** A QQ plot of the Mahalanobis distance and Chi-square quantile further identifies outliers with an extreme outlier at the 7th quantile corresponding to the 18th data variable in A.
(DOCX)

## Acknowledgments

The authors are grateful to Peggy Auinger, MS for her contribution to the statistical analyses and for her comments on earlier drafts of the paper, and to Victoria Zhang, PhD for her assistance with the clinical laboratory protocols used in the evaluation of PRL.

## Author Contributions

**Conceptualization:** David A. Paul, Ismat Shafiq, Zoë Williams, G. Edward Vates.

**Data curation:** David A. Paul, Alejandra Rodrigue, Sam Haber, Ricky Hoang, Ismat Shafiq.

**Formal analysis:** David A. Paul, Alejandra Rodrigue, Nicholas Contento, Ricky Hoang.

**Investigation:** David A. Paul, Alejandra Rodrigue, Zoë Williams, G. Edward Vates.

**Methodology:** David A. Paul, Ismat Shafiq, Zoë Williams, G. Edward Vates.

**Project administration:** Sam Haber.

**Supervision:** Ismat Shafiq, Zoë Williams, G. Edward Vates.

**Writing – original draft:** David A. Paul, Alejandra Rodrigue.

**Writing – review & editing:** David A. Paul, Nicholas Contento, Sam Haber, Redi Rahmani, Adnan Hirad, Ismat Shafiq, Zoë Williams, G. Edward Vates.

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
