## [Decision Letter · Decision Letter 0]

23 Feb 2022

PONE-D-21-39870Prolactin at moderately increased levels confers a neuroprotective effect in non-secreting pituitary macroadenomasPLOS ONE

Dear Dr. Paul,

Thank you for submitting your manuscript to PLOS ONE. After careful consideration, we feel that it has merit but does not fully meet PLOS ONE’s publication criteria as it currently stands. Therefore, we invite you to submit a revised version of the manuscript that addresses the points raised during the review process.

 Please carefully address all questions raised by the reviewer. The revised paper paper will be resent for review upon receipt.

We look forward to receiving your revised manuscript.

Kind regards,

Tudor C. Badea, M.D., M.A., Ph.D.

Academic Editor

PLOS ONE

Journal Requirements:

Reviewers' comments:

Reviewer's Responses to Questions

**Comments to the Author**

1. Is the manuscript technically sound, and do the data support the conclusions?

Reviewer #1: Partly

2. Has the statistical analysis been performed appropriately and rigorously? 

Reviewer #1: Yes

3. Have the authors made all data underlying the findings in their manuscript fully available?

Reviewer #1: Yes

4. Is the manuscript presented in an intelligible fashion and written in standard English?

Reviewer #1: Yes

5. Review Comments to the Author

Reviewer #1: The study by Paul et al. aims to investigate whether there is a causal relationship between serum prolactin (PRL) levels and the thickness of the retinal nerve fiber layer in patients that suffered optic chiasma compression associated with macroadenomas. This is interesting clinical study. The PRL field of research started decades ago, but in vivo studies are certainly missing to better understand its complex and intricated roles. In general, the study makes sense, data are a bit scarse, but it is inherent to the nature of retrospective studies. I consider that it is worth publishing in Plos One, but I have some concerns that need to be addressed before recommending acceptance with no revision. In particular, some aspects about the statistics, data analysis and presentation, and discussion need to be addressed.

Major comments:

- Even if several effects of PRL are now known to do not depend on the sex, main ones do depend on the sex. It is therefore necessary to clearly study this issue when one deals with PRL. Biological sex is not mentioned in the Materials and Methods section. P.7 lines 171-172: the authors state “There was no difference in male/female make-up between the PRO and Control group”. Which figure does support this data? Line 171 p.7, the “male/female” terms refer to animals. Please correct.

- page 9, lines 206-208: What is the rationale to eliminate what the authors consider as “extreme outliers”? Working with confidence intervals may be a good way to solve this type of issues. In this line, I did not find any estimate of sample size to reach statistically significant conclusions. Even if the study is retrospective, such estimate is necessary. At least to get an idea of the power of the statistical data.

- p.7, lines 175-176: what does this “Of the patients in the PRO group, 22 eyes had mean deviation measurements” mean? Similar comment for the following sentence “Of the Control group 51 eyes had mean deviation data available.”?

p.2 line 55: The authors state that “Of 52 non-secreting macroadenoma patients, 12 had moderate elevation of prolactin secondary to stalk effect”. Do the authors have direct evidence of that, I mean, for their own data? Please, clarify.

p.3 line 94-95: The authors state that “we investigate the relation between serum hormone levels of prolactin, RNFL thickness and visual function in a retrospective cohort of pituitary macroadenoma patients.” How visual function was studied?

- p.13, lines 289-291: The authors state that “While there was no observed increase in visual function at increasingly higher levels of prolactin in the PRO group, […]”, but I cannot find any data supporting that. RNFK thickness is not a measure of visual function.

p.7, line 181: What is the physiological relevance of the temporal quadrant of the retinal nerve fiber layer?

Please define the superior, nasal, temporal, and inferior quadrants in the Methods section.

- About the tumor characteristics, the authors state that “Tumor size was identified by a single trained investigator and the largest dimension in any direction was recorded” (p.5, lines 135-136). The tumor size should be measured. Where are they recorded?

- Please detail the PRL assay (p.5 line 139).

- p.5-6 (lines 145 and 149): the number of tested eyes in the control group does not match. Please correct.

- p.6, line 158: what are normal levels of serum PRL?

- Please discuss why a moderate increase in serum PRL levels is beneficial, while a greater increase is not, from a mechanistic point of view. In this line, p.13, lines 293-295, the authors mention that opposite actions of PRL may be due to the fact that PRL exerts both inhibitory and excitatory actions and that PRL regulates gonadotropin release. This seems to imply that the opposite actions of PRL on RFNL are due to the activation of opposite signaling pathways, which may happen, but what directs the resulting effect of PRL. Why at high doses, the neurotoxic effects would prevail and at moderate doses, the neuroprotective effects would be more important? It seems to me that this explanation does not take into account that the PRL receptors respond to their agonist with a bell-shape curve. I am referring to the fact that at high concentrations of PRL, PRL receptors desensitize. Conversely, small amount of agonist activates poorly the receptor. This may explain why the neuroprotective effect is lost if patients have too much or too little of PRL. In addition, PRL can be transformed into active peptides (vasoinhibins/16K prolactin), which also have neuronal effects. Please, complete the discussion to fully discuss this crucial issue for the study. At last, what are the actions of gonatropin on RNFL/neuronal survival?

- p. 12, lines 283-283: The authors mention that their control group is older than their PRL group. Then they discuss the implication of aging in RNFL thickness, but could they also include that PRL levels also decrease with age? It would reinforce their point that physiological PRL levels are neuroprotective and that extreme serum PRL levels, either too high or too low, are detrimental for neurons.

- p.4 lines 104-108: Please add references that support the two ways through which serum PRL can increase in humans.

- p.4 lines 11-112: I do not understand the logic behind the use of NS+ and NS in the control group… Does N stand for normal and S for… serum?

- Table 1: The p value for the age at diagnosis needs to be checked and aligned.

Minor comments:

- p.4, line 102: “[…] visual pathway structures, we can explore […]”. A coma seems to be missing.

- p.5, lines 126-130: The institution's reference sounds repetitive.

- It would be easier to detect the figures and tables in the Result section if they were written in bold.

- Please regroup all figure legends, including table legends at the end of the manuscript.

- p.9 lines 203: “Post-hoc” should be written in italics.

- p.10, line 238: Please correct typo “[…] our data demonstrate that […]”.

- p.11 line 260, please add a coma before but.

- p.13, lines 293-294: please add a coma after hypothalamus.

6. PLOS authors have the option to publish the peer review history of their article (what does this mean?). If published, this will include your full peer review and any attached files.

Reviewer #1: **Yes: **Stéphanie C. Thébault

---

## [Author Response · Author response to Decision Letter 0]

2 May 2022

Responses to Highlighted Reviewer’s Comments:

1. Even if several effects of PRL are now known to not depend on the sex, main ones do depend on the sex. It is therefore necessary to clearly study this issue when one deals with PRL. Biological sex is not mentioned in the Materials and Methods section. P.7 lines 171-172: the authors state “There was no difference in male/female make-up between the PRO and Control group”. Which figure does support this data? Line 171 p.7, the “male/female” terms refer to animals. Please correct

Response: We have included a statement acknowledging biological sex in the methods section. We also refer readers to Table 1, which includes demographic data on biological sex. Male and Female are used to describe biological sex as documented in the medical chart. 

p.5 lines 130-131

Both Male and Female patients were included in the study, as defined by their biological sex on chart review. 

p.8 lines 213-214

There was no significant difference in biological sex between patients in the PRO and Control groups (Table 1).

2. What is the rationale to eliminate what the authors consider as “extreme outliers”? Working with confidence intervals may be a good way to solve this type of issues. In this line, I did not find any estimate of sample size to reach statistically significant conclusions. Even if the study is retrospective, such estimate is necessary. At least to get an idea of the power of the statistical data.

Response: Re: Outlier analysis. We have provided further context for the use of this method within the body of the paper, emphasizing the how this is the most objective way to identify a potentially erroneous value. Additional post-hoc validation was also performed. 

p.10 lines 251-257

In plotting the mean deviation data, a single data point varied from the rest. Prior to running statistical analysis on the data set, Mahalanobis distance was used – which identified an extreme outlier in the mean deviation data for the NS+ group (See Supplemental Data). This was validated post-hoc using the extreme studentized deviate test (z-score 3.66, p <0.05). Mahalanobis distance and extreme studentized deviate test did not identify any other outliers in either the PRO or NS groups. The extreme outlier was removed from the mean deviation analysis.

Re: Sample size estimate and power analysis. A power analysis was performed based on data publicly available in the literature and added to the methods section of the manuscript. 

p.5 lines 140 -150

Power analysis: A power analysis was conducted based on the initial hypothesis that elevated levels of prolactin in prolactin-secreting macroadenoma patients will demonstrate attenuation of injury to the retinal nerve fiber layer compared with non-secreting macroadenoma patients. Previous research indicates that pre-operative mean RNFL thickness in pituitary macroadenoma patients (standard deviation) is 81.9 µm (8.8) (Moon et al.). We anticipated a similar mean RNFL thickness for non-secreting pituitary macroadenoma patients in this study. For patients with elevated prolactin, we anticipated a mean RNFL thickness to be closer to the control group mean of 91.9 µm (9.7), as reported in Moon et al. A total sample size of 30 participants (15 per group) was found to provide 80% power to detect a 10 µm difference in mean RNFL thickness between the two groups, using a Welch’s T-test and a 5% significance level.

References:

 Moon CH, Hwang SC, Kim BT, Ohn YH, Park TK. Visual prognostic value of optical coherence tomography and photopic negative response in chiasmal compression. Invest Ophthalmol Vis Sci. 2011;52(11):8527-33.

3. p.7, line 181: What is the physiological relevance of the temporal quadrant of the retinal nerve fiber layer? Please define the superior, nasal, temporal, and inferior quadrants in the Methods section.

Response: We have included additional text that better defines the quadrants, their structural/functional relationship to the visual field, and increased sensitivity of the temporal RNFL quadrant to chiasmal compression. 

p.6 lines 170-179

Measures of RNFL thickness were reported as a function of clock hour position of the fovea and subsequently grouped into anatomical quadrants based on a standard division of the visual field (e.g., for right eyes; superior quadrant: 11 and 1 o’clock; nasal quadrant: 2, 3, and 4 o’ clock; inferior quadrant: 5 and 7 o’ clock; and the temporal quadrant: 8, 9, and 10 o’clock). The 12 o’clock and 6 o’clock positions were excluded from analysis secondary to nontemporal overlap of retinal ganglion cell projections (Schneider et al.). The temporal RNFL quadrant demonstrates increased sensitivity to injury with pituitary tumors secondary to its association with crossing retinofugal fibers at the level of the optic chiasm (Moon et al.).

References:

 Schneider CL, Prentiss EK, Busza A, Matmati K, Matmati N, Williams ZR, et al. Survival of retinal ganglion cells after damage to the occipital lobe in humans is activity dependent. Proc Biol Sci. 2019;286(1897):20182733.

 Moon CH, Hwang SC, Kim BT, Ohn YH, Park TK. Visual prognostic value of optical coherence tomography and photopic negative response in chiasmal compression. Invest Ophthalmol Vis Sci. 2011;52(11):8527-33.

4. About the tumor characteristics, the authors state that “Tumor size was identified by a single trained investigator and the largest dimension in any direction was recorded” (p.5, lines 135-136). The tumor size should be measured. Where are they recorded?

Response: Tumor size is reported in Table 1. The text has been updated to better reflect this.

p.8 line 217-218 

There was no significant difference in tumor size between the PRO (22.1 mm ± 14.7) and control groups (25.1 ± 8.9; p-value = 0.20). See Table 1. 

5. Please detail the PRL assay (p.5 line 139).

Response: Additional text has been added to describe the PRL assay. 

p.6 lines 160-166

Briefly – antigen-specific monoclonal antibodies are coated onto beads and mixed with the PRL sample to allow an immune reaction to occur. Unbound sample is then washed away and a second monoclonal antibody with an electrochemiluminescent probe is added to the mixture to bind the PRL-antibody complex. An electrode is then introduced to the sample, which generates quantifiable electrochemiluminescence via an oxidation-reduction reaction that directly correlates to the amount of PRL present. 

6. p.5-6 (lines 145 and 149): the number of tested eyes in the control group does not match. Please correct

Response: We are very grateful to the reviewer for seeing this--we identified a typo in the submitted manuscript. The total number of eyes in the prolactinoma group is 27, and for the control group 104. The manuscript text has been updated to reflect this.

7. p.6, line 158: what are normal levels of serum PRL?

Response: The normal reference range for prolactin obtained in our laboratory is 4.8-23.3ng/ml.

p.6 line 159-160.

Serum prolactin was measured via Electrochemiluminescence Immunoassay (ELISA) with a reference range of 4.8-23.3ng/ml…

8. Please discuss why a moderate increase in serum PRL levels is beneficial, while a greater increase is not, from a mechanistic point of view. In this line, p.13, lines 293-295, the authors mention that opposite actions of PRL may be due to the fact that PRL exerts both inhibitory and excitatory actions and that PRL regulates gonadotropin release. This seems to imply that the opposite actions of PRL on RFNL are due to the activation of opposite signaling pathways, which may happen, but what directs the resulting effect of PRL. Why at high doses, the neurotoxic effects would prevail and at moderate doses, the neuroprotective effects would be more important? It seems to me that this explanation does not take into account that the PRL receptors respond to their agonist with a bell-shape curve. I am referring to the fact that at high concentrations of PRL, PRL receptors desensitize. Conversely, small amount of agonist activates poorly the receptor. This may explain why the neuroprotective effect is lost if patients have too much or too little of PRL. In addition, PRL can be transformed into active peptides (vasoinhibins/16K prolactin), which also have neuronal effects. Please, complete the discussion to fully discuss this crucial issue for the study. At last, what are the actions of gonadotropin on RNFL/neuronal survival?

Response: We appreciate the reviewer’s suggestion and have bolstered the discussion section to address these key issues – particularly with respect to proposed mechanisms that may be responsible for prolactin’s effect at moderate levels. With regards to the role of gonadotropin on RNFL/neuronal survival, we felt commenting on this would be beyond the scope of the current article. 

p.15 lines 370-380

Alternatively, chronic, extremely elevated levels of PRL in the PRO group can cause receptor desensitization and downregulation thus preventing the neuroprotective potential. The kinetics of the human prolactin receptor have shown it to behave with its agonist in a bell-shaped fashion implying supersaturation at high levels and decreased pathway activation at low level (Kinet et al.) Thus, the moderately elevated levels of PRL may work at peak PRL receptor activity without desensitization. Additionally, proteolytic cleavage of PRL generates a 16K PRL called vasoinhibin – which has been shown to have effects on vasculature by promoting vasopressin release, and on neurons by inhibiting neurite outgrowth (Castillo et al.). High levels of PRL would in turn result in increased levels of vasoinhibin, whose function may prevent or oppose the neuroprotective action of PRL.

References:

 Kinet S, Bernichtein S, Kelly PA, Martial JA, Goffin V. Biological properties of human prolactin analogs depend not only on global hormone affinity, but also on the relative affinities of both receptor binding sites. J Biol Chem. 1999;274(37):26033-43.

 Castillo X, Melo Z, Varela-Echavarria A, Tamariz E, Arona RM, Arnold E, et al. Vasoinhibin Suppresses the Neurotrophic Effects of VEGF and NGF in Newborn Rat Primary Sensory Neurons. Neuroendocrinology. 2018;106(3):221-33.

9. p. 12, lines 283-283: The authors mention that their control group is older than their PRL group. Then they discuss the implication of aging in RNFL thickness, but could they also include that PRL levels also decrease with age? It would reinforce their point that physiological PRL levels are neuroprotective and that extreme serum PRL levels, either too high or too low, are detrimental for neurons.

Response: We appreciate the reviewer’s suggestion to include this info and have added it to the discussion, along with relevant citations. 

p.14 lines 351-358

Additionally, a 24-hour PRL collection study found that older patients had a lower pulse mass and lower peak values of PRL secreted (Roelfsema et al.), although average PRL values overall did not significantly decline with age. Taken together, our data demonstrates that even with the added variable of age there is a preservation of retinal thickness in the group with mildly elevated prolactin. In other words, regardless of the natural retinal thinning and PRL changes that may come with age, older patients with moderately elevated PRL had greater RNFL thickness than older patients with significant hyperprolactinemia.

References:

 Roelfsema F, Pijl H, Keenan DM, Veldhuis JD. Prolactin secretion in healthy adults is determined by gender, age and body mass index. PLoS One. 2012;7(2):e31305.

10. p.4 lines 104-108: Please add references that support the two ways through which serum PRL can increase in humans.

Response: Two references have been added to support the mechanisms by which PRL can increase in humans.

 Serri O, Chik CL, Ur E, Ezzat S. Diagnosis and management of hyperprolactinemia. CMAJ. 2003;169(6):575-81.

 Mancini T, Casanueva FF, Giustina A. Hyperprolactinemia and prolactinomas. Endocrinol Metab Clin North Am. 2008;37(1):67-99, viii.

11. p.4 lines 11-112: I do not understand the logic behind the use of NS+ and NS in the control group… Does N stand for normal and S for… serum?

Response: NS stands for “non-secreting.” The text has been updated to better reflect this and to eliminate any confusion. NS+ represents “non-secreting” macroadenoma patients with an elevation of prolactin secondary to stalk effect. See p.4 lines 110-112. Thank you for this request to clarify. 

12. Table 1: The p value for the age at diagnosis needs to be checked and aligned.

Response: This value was missing a leading zero. This has been corrected. Thank you.

Responses to Minor Reviewer’s Comments:

p.4, line 102: “[…] visual pathway structures, we can explore […]”. A coma seems to be missing.

Corrected, thanks.

p.5, lines 126-130: The institution's reference sounds repetitive.

The institution reference has been removed. 

It would be easier to detect the figures and tables in the Result section if they were written in bold.

All references to figures are now in bold type.

Please regroup all figure legends, including table legends at the end of the manuscript.

Table Legends remain with the Tables in accordance with PLOS One Formatting guidelines.

p.9 lines 203: “Post-hoc” should be written in italics.

Corrected, thanks.

p.10, line 238: Please correct typo “[…] our data demonstrate that […]”.

Corrected, Thank you.

p.11 line 260, please add a coma before but.

Done.

p.13, lines 293-294: please add a coma after hypothalamus.

Done.

Responses to Editor’s Comments:

Please provide additional details regarding participant consent. In the ethics statement in the Methods and online submission information, please ensure that you have specified (1) whether consent was informed and (2) what type you obtained (for instance, written or verbal, and if verbal, how it was documented and witnessed). If your study included minors, state whether you obtained consent from parents or guardians. If the need for consent was waived by the ethics committee, please include this information.

We have addressed all editorial comments in the body of the manuscript. 

p.5 lines 136-138

The study protocol was approved by the institutional review board of the University of Rochester and the need for consent was waived. All data were anonymized before being accessed.

Please include captions for your Supporting Information files at the end of your manuscript, and update any in-text citations to match accordingly. Please see our Supporting Information guidelines for more information: http://journals.plos.org/plosone/s/supporting-information.

The Caption for the Supplemental Figure has been added at the end of the manuscript.

---

## [Decision Letter · Decision Letter 1]

16 May 2022

PONE-D-21-39870R1Prolactin at moderately increased levels confers a neuroprotective effect in non-secreting pituitary macroadenomasPLOS ONE

Dear Dr. Paul,

Thank you for submitting your manuscript to PLOS ONE. After careful consideration, we feel that it has merit but does not fully meet PLOS ONE’s publication criteria as it currently stands. Therefore, we invite you to submit a revised version of the manuscript that addresses the points raised during the review process.

It seems your answers have addressed the concerns of the reviewer. Please provide the additional information requested.

In particular, please describe not just the source of the antibody used, but the evidence demonstrating its specificity (western blot analysis, immunostaining of transfected cells, analysis in KO mice, etc.).

If that evidence is derived from other published papers, please provide the references to them.

We look forward to receiving your revised manuscript.

Kind regards,

Tudor C. Badea, M.D., M.A., Ph.D.

Academic Editor

PLOS ONE

Journal Requirements:

Additional Editor Comments (if provided):

It seems your answers have addressed the concerns of the reviewer. Please provide the additional information requested by the reviewer.

In particular, please describe not just the source of the antibody used, but the evidence demonstrating it specificity (western blot analysis, immunostaining of transfected cells, analysis in KO mice, etc.).

If that evidence is derived from other published papers, please provide the references to them.

Reviewers' comments:

Reviewer's Responses to Questions

**Comments to the Author**

1. If the authors have adequately addressed your comments raised in a previous round of review and you feel that this manuscript is now acceptable for publication, you may indicate that here to bypass the “Comments to the Author” section, enter your conflict of interest statement in the “Confidential to Editor” section, and submit your "Accept" recommendation.

Reviewer #1: (No Response)

2. Is the manuscript technically sound, and do the data support the conclusions?

Reviewer #1: Yes

3. Has the statistical analysis been performed appropriately and rigorously? 

Reviewer #1: Yes

4. Have the authors made all data underlying the findings in their manuscript fully available?

Reviewer #1: Yes

5. Is the manuscript presented in an intelligible fashion and written in standard English?

Reviewer #1: Yes

6. Review Comments to the Author

Reviewer #1: Overall all my comments have been answered satisfactorily and I think that the version improved.

I only noted a few details that need to be addressed:

Point 5: Please specify the antibody (manufacturer, ref.) used for the PRL assay.

Point 8: In the discussion about the potential mechanistic explanation for the opposite roles of PRL according to its levels, the "vasoinhibin" term should be used in plural. This was the whole point renaming them vasoinhibins instead of 16K prolatin, which infers that only 16 kDa N-terminal fragments are produced.

7. PLOS authors have the option to publish the peer review history of their article (what does this mean?). If published, this will include your full peer review and any attached files.

Reviewer #1: **Yes: **Stéphanie C. Thébault

---

## [Author Response · Author response to Decision Letter 1]

20 Jun 2022

Responses to Reviewer’s Comments:

1. Please specify the antibody (manufacturer, ref.) used for the PRL assay.

Additional Editor Comments (if provided): Please provide the additional information requested by the reviewer. In particular, please describe not just the source of the antibody used, but the evidence demonstrating it specificity (western blot analysis, immunostaining of transfected cells, analysis in KO mice, etc.). If that evidence is derived from other published papers, please provide the references to them.

Response: The manuscript has been updated to include the PRL assay manufacturer (Roche Diagnostics; Indianapolis, IN) and evidence of its sensitivity to detect prolactin (see new reference below, [16]). Of note, we have also included text to clarify that PRL testing was conducted in a CLIA certified clinical laboratory as part of the routine clinical workup for pituitary tumor patients at our institution. 

16. Fahie-Wilson M, Bieglmayer C, Kratzsch J, Nusbaumer C, Roth HJ, Zaninotto M, et al. Roche Elecsys Prolactin II assay: reactivity with macroprolactin compared with eight commercial assays for prolactin and determination of monomeric prolactin by precipitation with polyethylene glycol. Clin Lab. 2007;53(5-6):301-7.

Page 6, lines 149-156

Measurement of Serum Prolactin Levels: Serum prolactin was measured as part of the routine clinical workup for each patient via FDA approved Roche Elecsys Prolactin II Assay (Electrochemiluminescence Immunoassay [ECLIA]; Roche Diagnostics; Indianapolis, IN) with a reference range of 4.8-23.3ng/ml [15]. This assay demonstrates increased sensitivity to detect the concentration of monomeric prolactin by avoiding false elevation secondary to reactivity with macroprolactin [16]. Our clinical laboratory is a Clinical Laboratory Improvement Amendments (CLIA) certified laboratory and accredited by the Colleague of American Pathology and New York State Department of Health. 

2. In the discussion about the potential mechanistic explanation for the opposite roles of PRL according to its levels, the "vasoinhibin" term should be used in plural. This was the whole point renaming them vasoinhibins instead of 16K prolatin, which infers that only 16 kDa N-terminal fragments are produced.

Response: The text has been updated per the reviewers’ comments. 

Page 15, lines 346-350

Additionally, proteolytic cleavage of PRL generates active peptides (vasoinhibins/16K PRL) – which have been shown to have effects on vasculature by promoting vasopressin release, and on neurons by inhibiting neurite outgrowth [32]. High levels of PRL would in turn result in increased levels of vasoinhibins, whose function may prevent or oppose the neuroprotective action of PRL.

---

## [Editor Report · Decision Letter 2]

6 Jul 2022

Prolactin at moderately increased levels confers a neuroprotective effect in non-secreting pituitary macroadenomas

PONE-D-21-39870R2

Dear Dr. Paul,

We’re pleased to inform you that your manuscript has been judged scientifically suitable for publication and will be formally accepted for publication once it meets all outstanding technical requirements.

Kind regards,

Tudor C. Badea, M.D., M.A., Ph.D.

Academic Editor

PLOS ONE
---

## [Editor Report · Acceptance letter]

25 Jul 2022

PONE-D-21-39870R2 

Prolactin at moderately increased levels confers a neuroprotective effect in non-secreting pituitary macroadenomas 

Dear Dr. Paul:

I'm pleased to inform you that your manuscript has been deemed suitable for publication in PLOS ONE. Congratulations! Your manuscript is now with our production department. 

Kind regards, 

on behalf of

Dr. Tudor C. Badea 

Academic Editor

PLOS ONE